# Seroprevalence and Risk Factors for *Toxoplasma gondii* Infection in Sheep and Goats from Romania

Ionela Hotea *, Viorel Herman, Emil Tîrziu, Olimpia Colibar, Ileana Brudiu, Cătălin Sîrbu and Gheorghe Dărăbuș

Faculty of Veterinary Medicine, Banat University of Agricultural Sciences and Veterinary Medicine "King Michael I of Romania" from Timisoara, Calea Aradului, No. 119, Timisoara, 300645 Timis, Romania; viorelherman@usab-tm.ro (V.H.); emiltirziu@usab-tm.ro (E.T.); olimpiacolibar@usab-tm.ro (O.C.); brudiu@usab-tm.ro (I.B.); sirbucatalin90@gmail.com (C.S.); gheorghedarabus@usab-tm.ro (G.D.)
* Correspondence: ionelahotea@usab-tm.ro

**Abstract:** *Toxoplasma gondii* infections in sheep and goats are important economically and for public health for many countries, including Romania. In this study, seroprevalence and associated risk factors for *T. gondii* infection were determined in 2500 sheep and 500 goats from three counties in the Banat region of Romania. Antibodies to *T. gondii* were found in 1266 of 2500 (50.64%) sheep and in 375 (75%) of 500 goats, by using a commercial (IDEXX) ELISA. To evaluate the epidemiological status of the infection, some risk factors for ovine and caprine *T. gondii* infections were assessed. The main risk factors associated with *T. gondii* infections were breed, age, and the presence of cats in the farm. Seroprevalence was higher in animals older than three years and in purebred versus mixed breed animals. This is the first detailed study of risk factors associated with *T. gondii* seroprevalence in sheep and goats in Romania.

**Keywords:** *Toxoplasma gondii*; sheep; goats; risk factor

## 1. Introduction

*Toxoplasma gondii* infections are common in humans and animals worldwide. The ingestion of water and/or food, mainly undercooked meat, contaminated with oocysts, are the major sources of *T. gondii* infection in humans [1]. Sheep and goats are important sources of infection for humans worldwide, representing an important role in public health, as the consumption of infected milk or meat can facilitate the zoonotic transmission and ingestion of undercooked infected lamb is recognized as risk factor for *T. gondii* infection in Europe, especially pregnant women. Additionally, toxoplasmosis is an important cause of neonatal mortality in sheep and goats, resulting in reproductive and economic losses worldwide [2–5].

The number of sheep and goats in Romania is around 10 million heads [6]. In the Banat region, there are 1,201,135 sheep and 24,390 goats, according to the National Food Safety and Animal Health Authority [6]. Mutton is commonly consumed by Romania's citizens, and also exported to other European countries [7]. Thus, the produced meat may be a source of *T. gondii* infection for consumers both in Romania and abroad. Although *T. gondii* seroprevalence in sheep and goats has been previously reported in Romania [8–10], here we provide detailed epidemiological data, including the infection risk factors. Therefore, the objective of the present study is to present a seroepidemiological survey for *T. gondii* infection in sheep and goats raised in the Banat region of Romania. The importance of this study consists of identifying the main favorable factors that predispose small ruminants to *T. gondii* infection, and the results obtained to be working tools in the prevention and control of this parasitosis in sheep and goats.

## 2. Results

### 2.1. Toxoplasma Gondii Seroprevalence

Seropositive animals were found in all sheep and goat farms. Seroprevalence was different in those two species: 1266 of 2500 sheep (50.64%; 95% CI = 0.48–0.52) and 375 of 500 goats (75%; 95% CI = 0.71–0.78) were seropositive. Seropositivity in sheep varied among the 3 counties; 49.7% (95% CI = 0.47–0.52) in Timis county, 42.45% (95% CI = 0.37–0.47) in Arad county, and 61.33% (95% CI = 0.56–0.65) in Caras-Severin county. Comparing the two species, the seroprevalence was higher in goats (75%; 95% CI = 0.71–0.78) than in sheep (50.8%; 95% CI = 0.46–0.55).

### 2.2. Risk Factors

The seroprevalence of *T. gondii* was correlated with age, sex, and breed as risk factors (Tables 1 and 2). Seroprevalence was higher in adult (3–6 years) sheep (62.99%) and goats (93.28%) than in younger ones (30.06% and 51.15%, respectively). Higher seroprevalence was also recorded in males compared to females, the difference being more obvious in sheep (63.2% > 42.26%) than in goats (78.4% > 71.6%). Significant differences were also observed between small ruminant breeds, identifying a higher susceptibility in pure breed animals (58.74% in sheep and 82.74% in goats) than in mixed breeds (38.76% in sheep and 71.08% in goats).

The presence of cats on the farm was an important determinant factor; cats were present in over 90% of sheep farms (94%) and goats (90%). The presence of cats could be observed even in feed storage in 84% of sheep farms and in 70% of goat farms. The free movement of cats in the area also had consequences on the degree of contamination of drinking water for animals, identifying a higher seroprevalence (53.91% in sheep and 78.5% in goats) in farms that used surface water as a source of watering for sheep and goats. The use of the extensive and semi-extensive rearing system in Romania seems to be a favorable factor in transmitting *T. gondii* infection to sheep and goats (Table 2).

The logistic regression analysis indicated that age (3 to 6 years), breed, and cat access to grazing lands, feed storage facilities, and housing conditions were the most significant risk factors associated with *T. gondii* seroprevalence, both in sheep and goats (Tables S1 and S2).

**Table 1.** Risk factors associated with biological characteristics of animals.

| | Positive [1] | Negative [2] | % of Positivity | *p* * | CI ** 95% | OR *** | Positive [1] | Negative [2] | % of Positivity | *p* * | CI ** 95% | OR *** |
|---|---|---|---|---|---|---|---|---|---|---|---|---|
| | | | **SHEEP** | | | | | | **GOAT** | | | |
| **AGE (years)** | | | | | | | | | | | | |
| 1.3–3 | 282 | 656 | 30.06 | 0.00 | 0.24–0.34 | 0.29 | 111 | 106 | 51.15 | 0.00 | 0.04–0.13 | 0.08 |
| 3–6 | 984 | 578 | 62.99 | | 0.64–1.23 | 0.89 | 264 | 19 | 93.28 | | 0.07–0.22 | 0.13 |
| **GENDER** | | | | | | | | | | | | |
| Female | 634 | 866 | 42.26 | 0.00 | 0.36–0.50 | 0.43 | 179 | 71 | 71.6 | 0.080 | 0.46–1.04 | 0.69 |
| Male | 632 | 368 | 63.2 | | 1.99–2.76 | 2.35 | 196 | 54 | 78.4 | | 0.96–2.16 | 1.44 |
| **BREED** | | | | | | | | | | | | |
| Pure | 873 | 613 | 58.74 | 0.00 | 1.91–2.65 | 2.25 | 139 | 29 | 82.73 | 0.005 | 1.22–3.10 | 1.95 |
| Mixed | 393 | 621 | 38.75 | | 0.38–0.52 | 0.44 | 236 | 96 | 71.08 | | 0.32–0.82 | 0.51 |
| **TOTAL** | **1266** | **1234** | **50.64** | | | | **375** | **125** | **75.00** | | | |

[1]—Positive animals from total tested; [2]—Negative animals from total tested. * *p*-value; ** CI—confidence interval; *** OR—odds ratio.

**Table 2.** Risk factors associated with the rearing environment.

| Risk Factor | Positive [1] | Negative [2] | % of Positivity | *p** | CI ** 95% | OR *** | Positive [1] | Negative [2] | % of Positivity | *P* * | CI ** 95% | OR *** |
|---|---|---|---|---|---|---|---|---|---|---|---|---|
| | | | **SHEEP** | | | | | | **GOATS** | | | |
| | | | **Cat presence on pastures and in pens/barns** | | | | | | | | | |
| Yes | 1220 | 1130 | 51.91 | 0.000 | 1.71–3.49 | 2.44 | 352 | 98 | 78.22 | 0.000 | 2.32–7.68 | 4.22 |
| No | 46 | 104 | 30.66 | | 0.29–0.59 | 0.41 | 23 | 27 | 46.00 | | 0.13–0.43 | 0.24 |
| | | | **Cat presence in feed storage facilities** | | | | | | | | | |
| Yes | 1158 | 942 | 55.14 | 0.000 | 2.62–4.21 | 3.32 | 312 | 38 | 89.14 | 0.000 | 7.10–18.1 | 11.34 |
| No | 108 | 292 | 27.00 | | 0.24–0.38 | 0.30 | 63 | 87 | 42.00 | | 0.06–0.14 | 0.09 |
| | | | **Source of drinking water** | | | | | | | | | |
| Surface | 620 | 530 | 53.91 | 0.003 | 1.09–1.49 | 1.27 | 157 | 43 | 78.50 | 0.141 | 0.90–2.09 | 1.37 |
| Underground | 646 | 704 | 47.85 | | 0.67–0.92 | 0.78 | 218 | 82 | 72.66 | | 0.48–1.11 | 0.73 |
| | | | **Production system** | | | | | | | | | |
| Extensive | 161 | 139 | 53.66 | 0.264 | 0.90–1.46 | 1.15 | 71 | 29 | 71.00 | 0.303 | 0.47–1.26 | 0.77 |
| Semi-extensive | 1105 | 1095 | 50.22 | | 0.68–1.11 | 0.87 | 304 | 96 | 76.00 | | 0.79–2.11 | 1.29 |
| **TOTAL** | **1266** | **1234** | **50.64** | | | | **375** | **125** | **75.00** | | | |

[1]—Positive animals from total tested; [2]—Negative animals from total tested. * *p*-value; ** CI—confidence interval; ***OR—odds ratio.

### 3. Discussion

The median seroprevalence identified in small ruminants in the Banat region shows that more than half of the overall number of the study animals tested positive for the *T. gondii* infection. The distribution of ovine seropositivity in western Romania indicated that Caras-Severin county had the highest levels of *Toxoplasma* seroprevalence, followed by Timis county, with Arad county having the lowest seroprevalence. The median prevalence in sheep from the Banat region (50.64%) is comparable to the prevalence determined in central Romania in 2009 by Titilincu et al. (50%), using the IFAT technique [11]. The prevalence shown in our study is slightly higher than the one found in 2008 by Iovu et al. (45.7%) and in 2008 by Militaru et al. (27.85%) in sheep from southern Romania [12,13]. The levels of prevalence in western Romania are lower than those detected in central Romania by Titilincu et al. in two separate studies (64.34% and 64.9%, respectively) conducted in 2009, using the same commercial kit (Chekit Toxotest, Liebefeld-Bern, Switzerland) [11].

Relief and climate may account for the differences in prevalence in the three Romanian counties. The ability of sporulated oocysts to survive up to 12 months in favorable environmental conditions (temperature, humidity) is well-documented [1]. In this study the highest seroprevalence was found in Caras-Severin county, which has a predominantly mountainous relief and where the mean annual precipitation is over 1000 mm per square meter. A seroprevalence of approximately 50% was identified in Timis county, where the relief is mainly low and flat, with mean annual temperature higher than in the other two counties, but with higher mean annual precipitation. The lowest seroprevalence was determined in Arad county where all forms of relief are present, and where the mean annual temperature is close to those in the other two counties. However, the mean annual precipitation is lower, which indicates that seroprevalence of *T. gondii* infection and oocyst survival in the environment are influenced mainly by atmospheric humidity, though atmospheric temperature is also a relevant influencing factor. The correlation between climate factors and oocyst survival in the environment and therefore the incidence of *T. gondii* seroprevalence in sheep has been described by numerous specialists [14,15].

Seropositivity in goats was found to be higher than in sheep. The selective feeding habits of goats do not explain the high seroprevalence detected in this species. Prevalence identified in this study in goats from western Romania (75%) is significantly higher than in goats from central Romania, as reported in 2009 by Titilincu et al. (64.15%) and in 2012 by Iovu et al. (52.8%) [11,16].

The association of *Toxoplasma* seropositivity with biological characteristics of animals has shown that age, gender, and breed can influence ovine and caprine vulnerability to *T. gondii* infection. High values of seroprevalence both in sheep and in goats have been obtained in animals aged more than three years, in male animals, and in pure breeds. Seroprevalence increases with age, which is attributable to an increased cumulative chance of exposure to environmental contamination [1,17,18]. In Romania, a study in lambs revealed a low seroprevalence of 6.5% [19]. The gender of the animals may influence genetically their susceptibility to *Toxoplasma* infection. In our study, male sheep and goats had higher seroprevalence than females. In 2010, in Brazil, Lopez et al. identified a prevalence higher in rams (64%) than in ewes (31%) [20]. In Romanian literature, there is just one study that presents the seroprevalence in rams as being 65.7% [21].

Since small ruminants are exclusively herbivorous, the main source of contamination with *T. gondii* is the ingestion of oocysts shed in the environment by felids. The presence of other wild felids that could be shedding oocysts into the environment can be considered as a risk factor for small ruminants [1,7]. Thus, environmental factors associated with the infection are extremely important, especially because of the individual farming practices meet in the studied farms. The main risk factors identified were: direct or indirect contact with cat faeces, contaminated feeds, contaminated drinking water, and flock management practices. Analysis of epidemiological data revealed that on most farms, included in this study, cat activities are not monitored and are allowed access to pastures, pens, and barns. Oocyst shedding is thus widespread, which creates increased opportunity of exposure to *T.*

*gondii.* The access of cats to feed storage units further increases the risk of contamination due to the cats' habit of burying their faeces; these places are ideal for such unsavory feline activities. A small number of farmers declared that they did not own cats, but that does not ensure a parasite-free environment, since contamination can be caused by their neighbours' cats or by feral cats [22]. Feline presence near or on sheep and goats farms, classified as risk factor associated with *T. gondii* infection, has been studied extensively by other authors too, and increased seroprevalence levels have been found in small ruminants that have come in direct or indirect contact with cats [23,24]. The presence of abortions caused by *Toxoplasma gondii* in small ruminants, and the felines that can consume the aborted tissues with cysts of the parasite, can be also an important way of transmission [1].

Surface waters (rivers, lakes) used by animals may become sources of infestation if cats deposit their faeces in their vicinity. Not even underground sources of water (fountains, wells) are risk-free due to the widespread practice of leaving the water in the watering troughs for several days and simply filling them up as they empty. If this water is accessible to cats, it becomes another source of infestation, which has been proved by the negligible differences ($p > 0.001$) between the seroprevalence levels found in animals drinking surface water on the one hand, and those drinking underground water on the other. Vesco et al. (2007) suggest that the use of surface waters as a source of drinking water for small ruminants may be regarded as a risk factor for *T. gondii* infection [24].

Extensive and semi-extensive farming are the most used types of sheep and goats rearing in Romania. Increased seroprevalence in animals reared in this way confirms the fact that the free and unmonitored circulation of cats is the main risk factor associated with *Toxoplasma* infection in animals in our country. Other studies report similar seroprevalence in small ruminants raised in the extensive production system [25,26].

It is common knowledge that the prevalence of toxoplasmosis varies according to the degree of the population's awareness of the disease. In Romania, small ruminants are slaughtered at 5 or 6 years of age, close to the end of their economic rentability. High seroprevalence identified in animals older than 3 years increases the risk of infection for humans, especially pregnant women and people with weakened immune systems, the main route of transmission being ingestion of raw or undercooked meat/meat products and unpasteurized milk. A seroprevalence of 57.6% found in women of child-bearing age and of 55.8% in pregnant women in Timis county supports this assumption [27,28]. While cat presence on farms is difficult to control, personal hygiene and adequate cooking of meat ($67^0$C in the center of the piece of meat) can reduce the risk of infection [1,29].

Increased seroprevalence in small ruminants poses a risk to the cats themselves, which become infected after eating the placenta or the raw meat of seropositive animals, thus allowing the parasite to resume its life-cycle. This fact is highly relevant in relation to the high *Toxoplasma* seroprevalence levels in cats in the Arad and Timis counties (80.6% and 62.8%, respectively) [26,30], hence the need for the general population, and especially for animal breeders to be aware of the epidemiological data regarding this parasitic infection.

Public health is the goal of the ongoing research on *Toxoplasma* infection in farm animals, including small ruminants, from western Romania and on its zoonotic transmission to humans. This research will benefit veterinarians and animal breeders because it will enable them to take adequate steps to protect these species i.e. small ruminants, which are highly vulnerable to *T. gondii* infection. It will also help specialists involved in food safety programs and finally, human epidemiologists.

## 4. Materials and Methods

### 4.1. Study Area

In Romania, sheep and goats are mainly reared in extensively and semi-extensively system. The extensive system farming is based on the maintenance of sheep, especially in pastures and feeding them only green mass during the grazing period or with other occasional sources of feed. In the semi-extensive breeding system, the alternation between the maintenance of sheep in pastures and their maintenance in shelters is practiced, and the

feeding is based on green mass through grazing, fibrous fodder of culture, and cereals. This study was conducted on sheep and goats from 60 farms, in 50 locations, in three counties of the Banat region: Timis, Arad, Caras-Severin (Figure 1, Table 1).

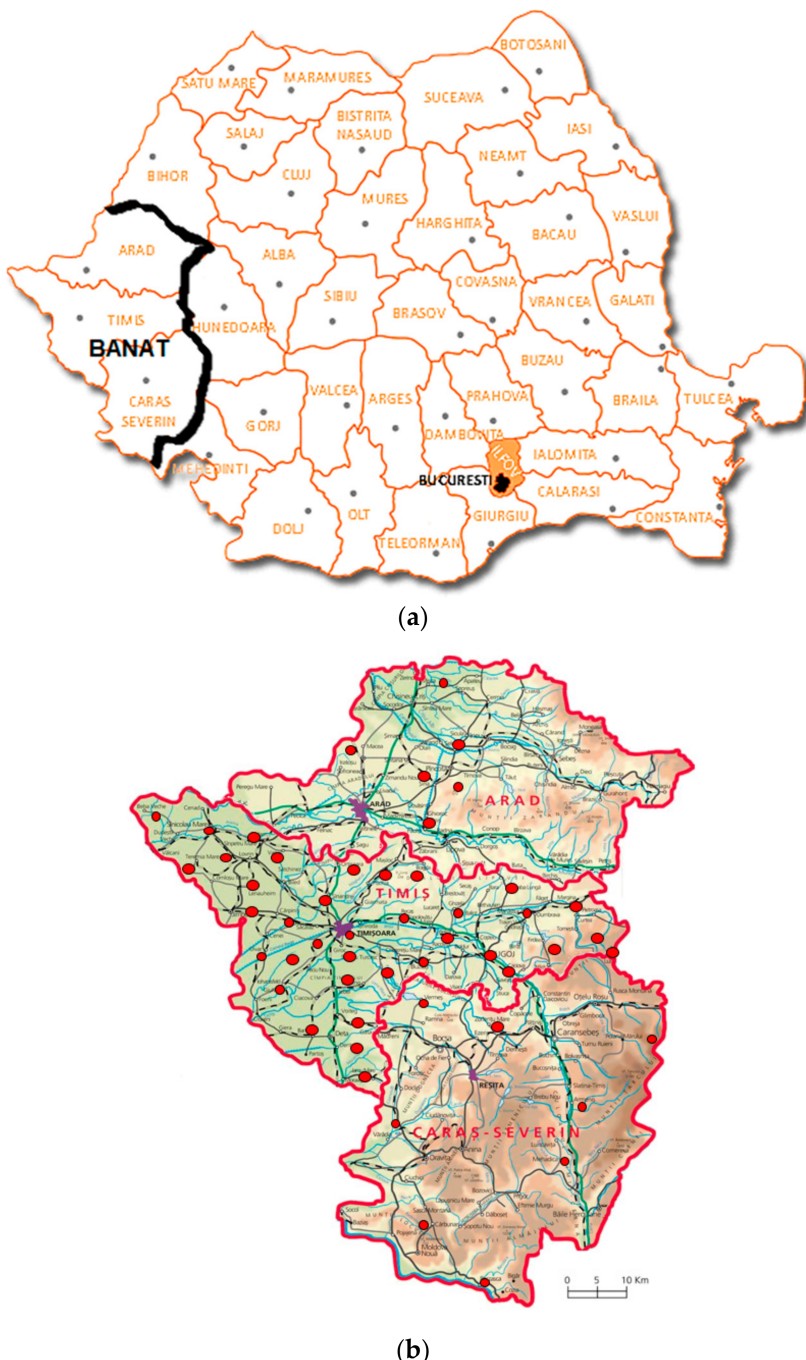

(**a**)

(**b**)

**Figure 1.** Study area (**a**) including 50 locations (**b**) in Banat Region, Romania.

Timis county has a predominantly low and flat terrain (relief), with a transitional climate with sub-Mediterranean influences. The mean annual temperature is 10.8 °C, and the mean annual precipitation is 873.6 mm per square meter. Arad county has a varied relief i.e. plains, mountainous hills, and low mountains, and the climate is moderate continental with sub-Mediterranean influences. The mean annual temperature is 10.5 °C and the annual precipitation averages 759.4 mm per square meter. In Caras–Severin county, the relief is mostly mountainous, with a moderate continental climate with sub- Mediterranean

influences. The mean annual temperature is 10.3 °C and annual precipitation is 1012.3 mm per square meter, on average [10].

### 4.2. Animals and Serum Collection

For this study, blood samples were collected during serological surveillance campaigns for sheep health. These campaigns are carried out at the national level, annually, in compliance with veterinary sanitary norms regarding the assurance of animals' welfare and avoidance of causing harm to animals. Thus, the collection of blood samples was carried out with the consent of the animal owners and in compliance with the national legislation in force regarding animal protection.

The study was conducted on 2500 sheep and 500 goats from 3 counties involving 50 farms in the Banat region, Romania (Table 3). From 10 farms, both sheep and goats were sampled. The animals, both male and female, were reared on privately owned farms. The age of the animals ranged from 1.3 to 6 years in sheep and 1.5 to 6 years in goats. They were pure breeds (Merinos, Țigaie, Țurcană—indigenous Romanian sheep breeds; Carpathian and Banat's White goat—indigenous Romanian goat breeds), as well as mixed breed, all animals being for mixed production—meat and milk. The flock holdings were 200 to 1700 sheep or 75 to 400 goats.

**Table 3.** Studied animals and their biological characteristics.

| SHEEP | No. Of Samples | Age 1.3–3 Years | Age 3–6 Years | Females | Males | Pure Breed | Mixed Breed |
|---|---|---|---|---|---|---|---|
| **Timis** | 1650 | 619 | 1031 | 900 | 750 | 981 | 668 |
| **Arad** | 400 | 151 | 248 | 350 | 50 | 237 | 162 |
| **Caras-Severin** | 450 | 168 | 283 | 350 | 100 | 268 | 184 |
| **TOTAL SHEEP** | **2500** | **938** | **1562** | **1500** | **1000** | **1486** | **1014** |
| **GOATS Timis** | **500** | **217** | **283** | **250** | **250** | **168** | **332** |

Blood samples (5 mL each) were collected by a jugular venipuncture in vacutainers, from 50 animals from each flock. After separating the sera, the serum samples were stored at −20 °C until tested in the Parasitology and Parasitic Diseases Laboratory of the Faculty of Veterinary Medicine in Timisoara.

### 4.3. Serology

The indirect ELISA method (CHEKIT TOXOTEST—IDEXX Laboratories, Switzerland, Liebefeld-Bern) was employed to detect IgG antibodies to *T. gondii* infection in the serum samples. This kit has been previously used by others to test sheep sera [9]. The 96-well plate is coated with the *T. gondii* antigen, and the antigen-antibody complex forms when a peroxidase conjugate is added. The addition of the conjugate followed by the addition of the substrate will cause the positive samples to turn blue and, subsequently, with the addition of the stop solution, they will turn yellow. The microplate was read by a Tecan reader, and based on optical density readings (OD), the S/P values were calculated. The interpretation of the results was based on these values: percentage values under 30% was considered negative; percentage values of 30 to 100% or higher than 100% were classified as positive.

### 4.4. Risk Factors

In order to correlate the incidence of *T. gondii* infection with potential risk factors, two types of risk factors were tested: firstly, those related to biological characteristics of the animals (age, sex, and breed) (Table 3) and secondly, those related to rearing conditions (cat contact and cat access to grazing pastures, feed storage facilities, barns and pens; water sources, and type of animal husbandry) (see Table 2).

*4.5. Statistical Analysis*

The Minitab 15 and EpiInfo 7 programmes were used to perform the statistical analysis of the data. There were calculated values of $p$ ($\leq 0.05$), which highlight the relevant differences in seroprevalence values. The association of risk factors with *T. gondii* infection was done by logistic regression, by calculating the confidence interval (CI 95%) and the likelihood of the occurrence of the association (odds ratio, OR).

## 5. Conclusions

This study brings news regarding the epidemiology of *T. gondii* infection in small ruminants. The seroprevalence of toxoplasmic infection in Romania is increased both in sheep, but especially in goats. Adulthood, male sex, and pure breed are the individual factors that predispose small ruminants to *T. gondii* infection. The uncontrolled movement of cats on pastures, in animal shelters, as well as in feed storage, contributes massively to the spread of oocysts and to the increase of the oral infection rate of farm animals. The rearing systems in Romania seem to be favorable factors for the transmission of this parasite. Highlighting the high seroprevalence and risk factors in the transmission of toxoplasmosis in small ruminants in this study may be a starting point in raising farmers' awareness of the importance of the problem and the needs to implement control measures to prevent the transmission of *Toxoplasma gondii*.

**Supplementary Materials:** The following are available as supplementary materials https://www.mdpi.com/2673-6772/1/2/5/s1: Table S1. *T. gondii* seroprevalence in the studied sheep farms; Table S2. *T. gondii* seroprevalence in the studied farms, from which both sheep and goats were sampled.

**Author Contributions:** I.H., conceptualization, methodology, writing—original draft preparation, revision; V.H. and E.T., resources; O.C., writing—original draft preparation; I.B., statistical analysis; C.S., investigation; G.D. conceptualization, supervision. All authors have read and agreed to the published version of the manuscript.

**Funding:** This research was funded by UEFISCDI (CNCSIS), grant number PNCD II Nr. 51-013.

**Institutional Review Board Statement:** The study was conducted according to the guidelines of the Declaration of Helsinki, and approved by the Institutional Review Board of Banat University of Agricultural Sciences and Veterinary Medicine "King Michael I of Romania" from Timisoara, Faculty of Veterinary Medicine.

**Informed Consent Statement:** Not applicable.

**Data Availability Statement:** The data presented in this study are available in the Supplementary Materials.

**Acknowledgments:** Many thanks to Dubey JP for technical support.

**Conflicts of Interest:** The authors declare no conflict of interest.

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
