# Peer review of "Seroprevalence and Risk Factors for Toxoplasma gondii Infection in Sheep and Goats from Romania"

_parasitologia, doi:10.3390/parasitologia1020005_

Round 1

Reviewer 1 Report

INTRODUCTION

The paragraph below must be rewritten. Information must be maintained, but they are very repetitive. It can be summarized.

The number of sheep and goats in Romania is the fourth largest in Europe: 9.3 million

33 heads on December the 1st, 2007 [4]. In the Banat region, the size of ovine and caprine

34 livestock is significant: 1122096 sheep and 26072 goats in 2009, according to the National

35 Food Safety and Animal Health Authority [4]. Mutton raised in Romania is commonly

36 consumed by humans in Romania, and also exported to other European countries [4].

37 Consequently, the meat produced may be a source of T. gondii infection for consumers

38 both in Romania and abroad. Although T. gondii seroprevalence in sheep and goats has

39 been previously reported in Romania [5–7], here we provide detailed epidemiological

40 data and the risk factors were detailed so, the objective of the present study is to present

41 a seroepidemiological survey for T. gondii infection in sheep and goats raised in Banat

42 region of Romania. The importance of this study consists in identifying the main favorable

43 factors that predispose small ruminants to T. gondii infection, and the results obtained can

44 be working tools in the prevention and control of this parasitosis in sheep and goats.

RESULTS

(93.28%) than in younger ones (30.06% and  51.15%, respectively ). Higher seroprevalence was

(93.28%) than in younger ones (30.06%, respectively 51.15%). Higher seroprevalence was

DISCUSSION

review this paragraph. Confused

mental factors associated with the infection are extremely important, especially because

128 of the close links between the incidence Toxoplasma seroprevalence and individual farm

129 ing practices. The main risk factors

Change incidence – use prevalence (incidence menos new cases)

It is common knowledge that the incidence of toxoplasmosis varies according to the

157 degree of the population’s awareness of the disease.

Add a map with the localization of the municipalities or region Where the study was carried out.

Reviewer 2 Report

The manuscript reports an interesting issue that is of great interest due to the limited number of studies like this one in Romania, and because of the importance of the apicomplexan parasite Toxoplasma gondii for both animal and human health. 

In this study, seroprevalence and associated risk factors for T. gondii infection were determined in 2500 sheep and 500 goats from three counties in the Banat region of Romania.

Overall the paper is clear and well written, but I have found a small number of suggested changes listed below:

Line 16: "..infections were: breed, age and the presence...", delete ":"

Line 17: "Seroprevalences were..." change to "seroprevalence was..."

Lines 33-34: Update reference 4, this reference is for the years 2007/2009; I think that in more than 10 years this data has changed.

Line 46: Change "T. gondii" to "Toxoplasma gondii"

This section could be completed with a table indicating the seroprevalence on each farm, which would help to interpret the data clearly.

Lines 53-54: Include seroprevalence in sheep.

Line 58: Change "30.06%, respectively 51.15%" to "30.06% and 51.15% respectively"

Tables: Both tables are hard to read at first view, please modify the tables to facilitate their visualization. Please review all confidence intervals.

Lines 85-92: The authors compare seroprevalences between different techniques. The results obtained with different techniques are hardly comparable, please note here that although the prevalences are similar, it is necessary to be careful in the comparison and interpretation of the results between two different serological techniques.

Lines 109-113: "Seropositivity in goats was found to be higher than in sheep...". The authors indicate that goats are more selective in their feeding, and also their diet is mainly arbustive, which hinders the ingestion of oocysts that may remain in the vegetation. However, seroprevalence in goats is higher. Could the serological technique used in this work have something to do with this?

Line 126: change "cats" to "felids"

Lines 126-141: The discussion does not consider the presence of other wild felids that could be shedding oocysts into the environment. 

The discussion also does not consider the presence of abortions caused by Toxoplasma gondii in small ruminants, which felines feed on and consume the tissue cysts of the parasite.

Study area

It would help significantly if the authors incorporated a figure (map) with the locations of the study.

Animals and serum collection

A basic description of each farm, including number of animals, type of feed, presence of cats, type of production, among others, would help to interpret the results adequately.

Reviewer 3 Report

The manuscript "Seroprevalence and risk factors for Toxoplasma gondii infection in sheep and goats from Romania" reports the seroprevalence of Toxoplasma gondii infection in sheep and goats in different regions of Romania. In addition, some risk factors were determined to evaluate the epidemiological status of this infection in the regions studied, and to provide a tool for the prevention and control of this parasitic disease.

My comments:

- Manuscript elaboration. There are some grammar and punctuation mistakes that must be revised. Besides, some sentences may be rephrased to make them easier to understand (some of them are too long). Therefore, English must be revised. Here are shown some examples: line 14-15 (comma); line 26-29 (rephrase the sentence please. It is too long and confusing); line 33 (“on December the 1st”); line 58 (“…younger ones (30.06%, respectively 51.15%).” There must be a mistake in the parenthesis); among others through the manuscript.

- Line 25. Did authors mean “contaminated meat”?

- Line 27. “infected milk” Please, If you are sure about this affirmation, please could you add references to support that?

- Line 29. What about immunocompromised people? Would you consider this group of people in this sentence? You did mention them on line 160.

- Line 48. “(50.64%; 95% CI = 0.48 – 0.52)” Move parenthesis after “2500 sheep”.

- Line 54. Please, also add the percentage for sheep in this sentence (as you did for goats).

- Table 1. Please, indicates in table legends what does “p, CI 95%, OR” mean. It makes understanding easier for the readers. Same for table 2.

- Line 87. “IFAT technique” Please, write what IFAT means. It is the first time this appears on the text.

- Line 120-121. Add a reference for this sentence, please.

- Line 236. Which p values did you consider significant? Please, indicate on the text.

Round 2

Reviewer 2 Report

Dear authors, thanks for the revised manuscript, however some aspects have not been resolved correctly or adequately addressed in their new revised version of the manuscript.

Point 5: - This section could be completed with a table indicating the seroprevalence on each farm, which would help to interpret the data clearly.

Point 8: - Tables: Both tables are hard to read at first view, please modify the tables to facilitate their visualization. Please review all confidence intervals. Also, some prevalence and confidence intervals are not correct. Please check/correct invalid values.

Point 12: - Lines 126-141: The discussion does not consider the presence of other wild felids that could be shedding oocysts into the environment.

Point 13: - The discussion also does not consider the presence of abortions caused by Toxoplasma gondii in small ruminants, which felines feed on and consume the tissue cysts of the parasite.

These two points (12 and 13) have not been mentioned in the discussion, they are two limitations to the article, a paragraph about these aspects in the discussion would give more quality to the article.
